# A new test to detect impairments of sequential visuospatial memory due to lesions of the temporal lobe

**Thomas Eggert\*, Phuong Van Nguyen, Katharina Ernst, Sandra V. Loosli, Andreas Straube**

Department of Neurology, University Hospital, LMU Munich, Munich, Germany

\* eggert@lrz.uni-muenchen.de

## Abstract

This study investigates visuospatial memory in patients with unilateral lesions of the temporal lobe and the hippocampus resulting from surgery to treat drug-resistant epilepsy. To detect impairments of visuospatial memory in these individuals, a memory test should be specific to episodic memory, the type of memory in which the hippocampus is crucially involved. However, most known visuospatial memory tests do not focus on episodic memory. We hypothesized that a new sequential visuospatial memory test, which has been previously developed and applied only in healthy subjects, might be suitable to fill this gap. The test requires the subject to reproduce a memorized sequence of target locations in ordered recall by typing on a blank graphics tablet. The length of the memorized sequence extended successively after repeated presentation of a sequence of 20 target positions. The test was done twice on day one and again after one week. Visual working memory was tested with the Corsi block-tapping task. The performance in the new test was also related to the performance of the patients in the standard test battery of the neuropsychological examination in the clinical context. Thirteen patients and 14 controls participated. Patients showed reduced learning speed in the new sequential visuospatial memory task. Right-sided lesions induced stronger impairments than left-sided lesions. After one week, retention was reduced in the patients with left-sided lesions. The performance of the patients in commonly used tests of the neuropsychological standard battery did not differ compared to healthy subjects, whereas the new test allowed discrimination between patients and controls at a high correct-decision rate of 0.89. The Corsi block-span of the patients was slightly shorter than that of the controls. The results suggest that the new test provides a specific investigation of episodic visuospatial memory. Hemispheric asymmetries were consistent with the general hypothesis of right hemispheric dominance in visuospatial processing.

## Introduction

The hippocampus is known to play an essential role in the formation of episodic memory [1, 2]. Lesions of the hippocampus resulting from resection in temporal lobe epilepsy (TLE) are

**Data Availability Statement:** All relevant data are within the manuscript and its Supporting Information files.

**Funding:** The author(s) received no specific funding for this work.

**Competing interests:** The authors have declared that no competing interests exist.

typically linked with impairments in the acquisition of episodic memory [3]. Lesions of the left hippocampus are associated primarily with verbal memory [4] and lesions of the right hippocampus with visuospatial memory [5, 6]. These impairments concern the acquisition of a long-term memory rather than working memory, because both verbal and visuospatial working memory span can be preserved even with bilateral lesions of the hippocampus [7]. Clinical examinations of long-term memory usually focus on verbal memory tasks such as the California verbal learning test (CVLT) [8, 9] or the Word List task from the Consortium to Establish a Registry for Alzheimer's disease Neuropsychological Assessment Battery (CERADNAB) which assess similar aspects of memory [10].

Two commonly used tests for visual memory are the Rey-Osterrieth Complex Figure Test (ROCFT) [11, 12] and the "Diagnostikum für Cerebralschädigung" (DCS) [13, 14]. With the DCS, nonresected TLE-patients show impaired learning capacity more so in right-sided TLE than in left-sided TLE [15]. This impairment in the DCS does not seem to reflect a deficit of long-term memory only because Helmstaedter, Pohl [15] observed a similar hemispheric asymmetry of the relative impairments with respect to controls for both immediate and trained recall. The performance of nonresected TLE-patients in the ROCFT seems more difficult to interpret. Loring, Lee [16] reported that the ROCFT, "based solely on pass/fail performance is unable to discriminate between left and right seizure onset subjects". These visual memory tests focus on the memory of visual gestalt and not on episodic, sequential memory. This focus seems to be widespread in the literature on nonverbal memory. Moye [17] listed 32 different nonverbal memory tests, only 11 of which tested for delayed recall and all these tests required the subject to reproduce or to identify visual forms rather than sequences of spatial locations.

We found only a single study [18] that investigated sequential visuospatial memory. In this study, the training stimulus consisted of a number ($N_t$) of sequentially presented targets (one at a time, and each for two seconds). The targets were red solid dots that appeared on a subset of $2 \cdot N_t$ open circles that were permanently present. After the presentation of all targets, subjects were asked to place red wooden chips on those open circles where the targets had been shown. A maximum of 15 such combined training/reproduction trials were performed with the same target sequence. Tucker, Novelly [18] compared the reproduction performance in their sequential presentation mode with a training where all $N_t$ targets were presented simultaneously for 5 seconds. Examining a group of 24 TLE-patients who underwent unilateral lobectomy, Tucker, Novelly [18] found that patients with right-sided lesions were significantly more impaired in the sequential as compared to the simultaneous training mode. This hemispheric asymmetry of the impairment reached significance for the reproduction performance after the 15th training trial but not for the immediate recall (after the first trial). Thus, in contrast to the memory impairment in the DCS reported by Helmstaedter, Pohl [15], the hemispheric asymmetry of the patient's impairment in the sequential presentation mode affected the trained but not the immediate recall. Since the reproduction task of Tucker, Novelly [18] did not explicitly ask for ordered recall, it is not clear whether their memory task actually led to the successive buildup of sequential visuospatial memory. However, if so, the results would suggest that the right temporal lobe plays a specific role for this buildup of sequential visuomotor memory.

A sequential visuospatial memory is needed, for example, in landmark-based navigation, when a long sequence of visual cues is used to reconstruct a spatial path without representing its entire shape. Because of the apparent lack of sequential tasks on visuospatial memory, we developed in previous studies [19–21] a test for deferred imitation of long spatial sequences (DILSS) by either manual or ocular pointing movements. In this task, subjects acquire a very long sequence of target positions in a repeated succession of presentation and reproduction. It is described in more detail in the Methods section. Here we only briefly describe its

characteristic features and the results we obtained with it in our previous studies: 1) the length of the sequence (here 20) exceeds the capacity of the visuospatial working memory; 2) the spatial locations are continuously distributed and not constrained to a limited set as in the task of Tucker, Novelly [18]; 3) during sequence presentation, subjects are not allowed to move the motor effector (eye or hand); 4) the recall which alternates with sequence presentation has to be performed on a blank screen without intermediate feedback (a major difference from the serial reaction-time task); 5) the timing of the recall is not accelerated or constrained by an external trigger, but under the free control of the subject. Previous studies showed that the recall of sequences learned in this task is independent of the motor effector (pointing with eye or hand) and is retained for several days to weeks [20]. Compared to the serial reaction-time task with accelerated responses [22], DILSS shows a general lack of memory interference [21] and does not show typical features of implicit motor learning such as chunking [23] or error propagation between subsequent elements [24]. These findings suggest that the DILSS-task is suitable to examine the development of long-term visuospatial memory, which is explicit because during the presentation, subjects deliberately focus on the few (two to three) targets that are appended to the memorized sequence in the current trial. This awareness during encoding is a major characteristic of explicit memory [25]. During recall, subjects had ample time to imagine the target moving from one position to the next and to remember the spatio-temporal sequence in its so-called autonoetic awareness, a feature which is a conceptual characteristic of episodic memory [26]. Moreover, it seems difficult to verbally encode the memory content in the DILSS-task because the target positions are continuously distributed across the visual screen. Therefore, we hypothesize that the DILSS task tests a specific type of visuospatial memory that is sequential, explicit, and episodic.

In the present study, we applied the DILSS-task in TLE-patients with unilateral surgical resections of the hippocampus and the temporal lobe. The main motivation of the study is to test our hypothesis that the DILSS-task provides a test procedure specific for episodic, sequential visuospatial memory. Because of the generally accepted role of the hippocampus and the temporal lobe in episodic visuospatial memory [1, 2], this hypothesis predicts that learning performance in DILSS should be critically impaired by lesions of the temporal lobe. Following up on the study by Tucker, Novelly [18], the current study also further pursues the hemispheric asymmetry of the temporal lobe with respect to its role in sequential visuospatial memory. To cover a larger range of retention periods than in the standard clinical tests, we adopted a study design similar to that of Visser, Forn [27] with initial repetitive training and retention tests after 30 min and after one week. All experiments were performed long time after surgery.

## Methods

### Subjects

14 healthy controls (6 males; 8 females; age = 29.5±9.7 yrs) and 13 resected and seizure-free TLE-patients (8 males; 5 females, age = 40.2±11.1 yrs) participated in the study. Patients were recruited from the regular follow-up examinations of surgically treated TLE patients at the Department of Neurology of the University of Munich. All patients seen between 7/2018 and 7/2019 who were seizure free and agreed to participate in the study were included. The clinical characteristics of the patients are shown in Table 1. The onset of epilepsy was in adolescence or adulthood in all but one patient (Table 1, ID: 12; onset age: 8 yrs). Handedness was assessed using the Edinburgh Inventory [28]. Of the 14 control subjects, 2 were left-handed and 12 were right-handed. All patients were right-handed. Language dominance was not determined separately, since 96% of healthy right-handers are known to have left-hemispheric speech dominance [29]. 6 patients were operated on the left side and 7 were operated on the right

**Table 1. Clinical characteristics of the patients.**

| ID[a] | Gender | Age (yrs)[b] | First diagnosis (yrs)[c] | Surgery (yrs)[d] | AFS[e] | Pathology[f] | Surgery | # of surgeries | Engel[g] | Anti epileptic drugs | Employed |
|---|---|---|---|---|---|---|---|---|---|---|---|
| 5 | f | 35 | 20 | 14 | L | hippo-campal sclerosis | amygdalo-hippocam-pectomy | 1 | 1A | no | yes |
| 6 | m | 32 | 13 | 12 | R | DNET degree I | tumor resection | 1 | 1A | yes | yes |
| 9 | m | 33 | 11 | 10 | R | DNET degree I | tumor resection | 1 | 1A | no | yes |
| 12 | f | 45 | 37 | 9 | R | gangli-oglioma | amygdalo-hippocam-pectomy | 1 | 1A | yes | yes |
| 14 | f | 63 | 48 | 18 | L | hippo-campal sclerosis | anterior temporal lobe resection | 1 | 1A | yes | yes |
| 18 | m | 40 | 21 | 19 | R | hippo-campal sclerosis | amygdalo-hippocam-pectomy | 1 | 1A | no | no |
| 20 | f | 54 | 17 | 8 | R | hippo-campal sclerosis | amygdalo-hippocam-pectomy | 1 | 1A | no | yes |
| 22 | m | 40 | 11 | 10 | L | limbic encephalitis | amygdalo-hippocam-pectomy | 1 | 1A | yes | yes |
| 24 | m | 32 | 8 | 7 | L | hippo-campal sclerosis | amygdalo-hippocam-pectomy | 1 | 1A | no | yes |
| 27 | m | 52 | 20 | 14 | R | hippo-campal sclerosis | anterior temporal lobe resection | 1 | 1A | yes | yes |
| 28 | f | 27 | 9 | 1 | L | oligoden-droglial hyperplasia | anterior temporal lobe resection | 2 | 1C | yes | yes |
| 29 | m | 44 | 4 | 3 | R | focal cortical dysplasia | amygdalo-hippocam-pectomy | 1 | 1A | no | yes |
| 30 | m | 26 | 4 | 1 | R | hippo-campal sclerosis | amygdalo-hippocam-pectomy | 2 | 1A | yes | yes |

[a]*ID*: subject identifier.

[b]*Gender*: f: female, m: male.

[c]*Age*: age of the subject at the time of the experiment.

[d]*First diagnosis*: Time between first diagnosis and experiment.

[e]*Surgery*: Time between last surgery and experiment.

[f]*AFS*: affected side.

[g]*Engel*: Engel Epilepsy Surgery Outcome Scale: 1A: Completely seizure-free since surgery, 1C: Free of disabling seizures for at least 2 years.

side. The hippocampus was resected in all patients. All patients showed very good outcome of the surgery, as shown by the good ranking in the Engel Epilepsy Surgery Outcome Scale (minimally 1C) and by the observation that all but one able to cope with the demands of professional life (see column *Employed*). Tomographic images (MRT/CT) of 11 of the 13 patients taken post-operatively are shown in Fig 1. The study was approved by the ethics committee of the medical faculty of the LMU (project number: 18–400) and all subjects gave their written informed consent before the test.

## Setup

Subjects were seated in front of a graphics tablet (Cintiq 22HD, 60 Hz; size: 47.9 x 27.1 cm; resolution: 1920 x 1080 pixel; Wacom; Kazo, Saitama, Japan). The recording of the position and the pressure of a graphic stylus were event-triggered and reached sampling rates of more than 100 Hz during pen movements. The stylus was lifted between pointing movements, the locations of which were extracted offline by detecting the time points of the pressure peak. The surface of the tablet was adjusted so that it was nearly orthogonal to the viewing direction. The viewing distance was about 45 cm.

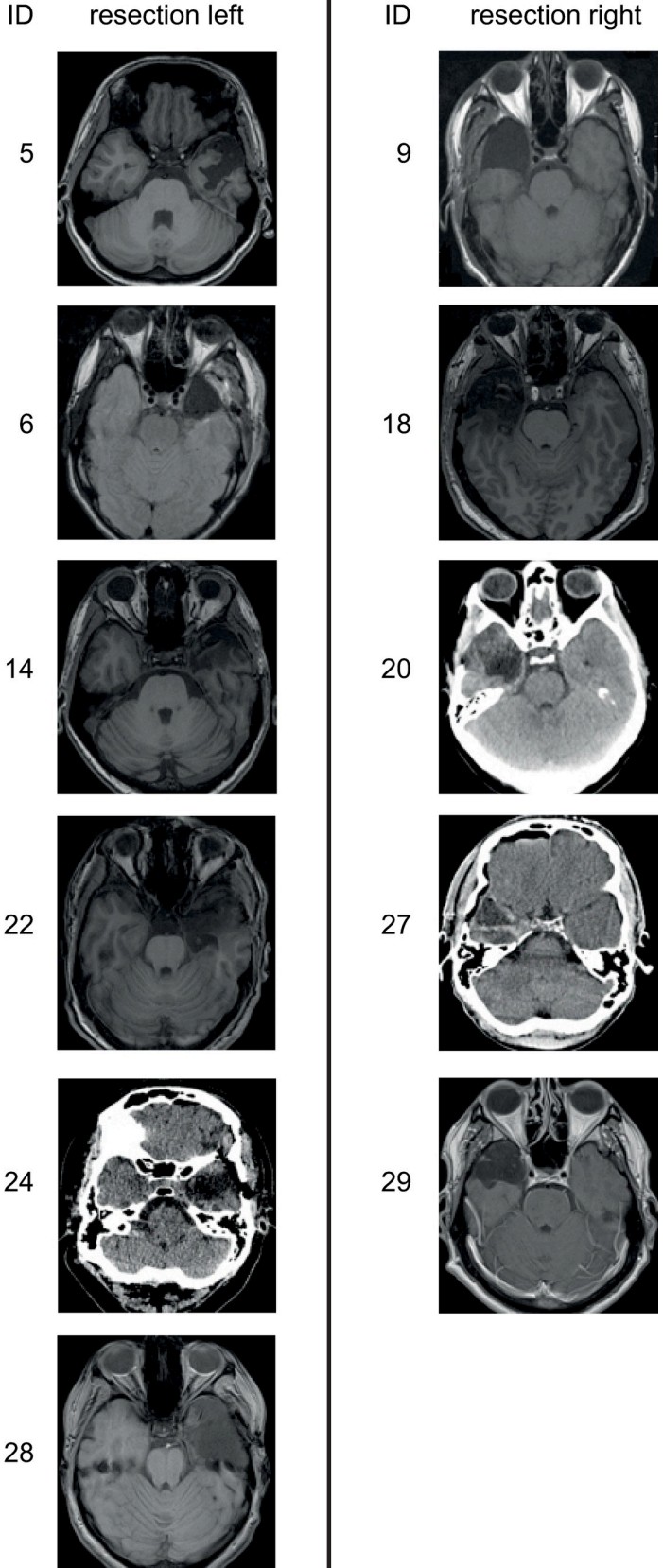

**Fig 1. MRI and CT slice images (axial) of 11 patients showing the temporal lobe resection.** To illustrate the size of the resections, the slices are centered on their largest extent, not on the hippocampus. The hippocampus was resected in all patients. No imaging data was available from two patients. The MRI of patient 28 was taken after the second surgery.

## DILSS-task

The DILSS-task [19–21] was performed in trials consisting of a presentation-phase during which the subject only observed the target on the screen, followed by a reproduction-phase during which the subject pointed with the graphic pen to the memorized target locations on the blank screen. A full training session consisted of 25 repetitions of these presentation-reproduction pairs. All instructions were given by the same examiner.

**Properties of the spatial sequence to be learned.** The spatial sequence consisted of 20 target locations, the x- and y-coordinates of which were equally distributed within a square with a width of 26 cm. The distribution of the targets was constrained in that no second target was allowed within a circle of 3.7 cm around any target, and no more than two targets were allowed within a circle of 5.6 cm around any target. The location of the last target of the sequence was identical with that of the start-fixation target. All subjects had to learn the same target sequence (Fig 2), which was generated once before the experiments.

**Presentation phase.** All targets were white crosses (width: 0.6 cm), presented on a homogeneous gray background. The presentation phase started with a fixation target, followed by

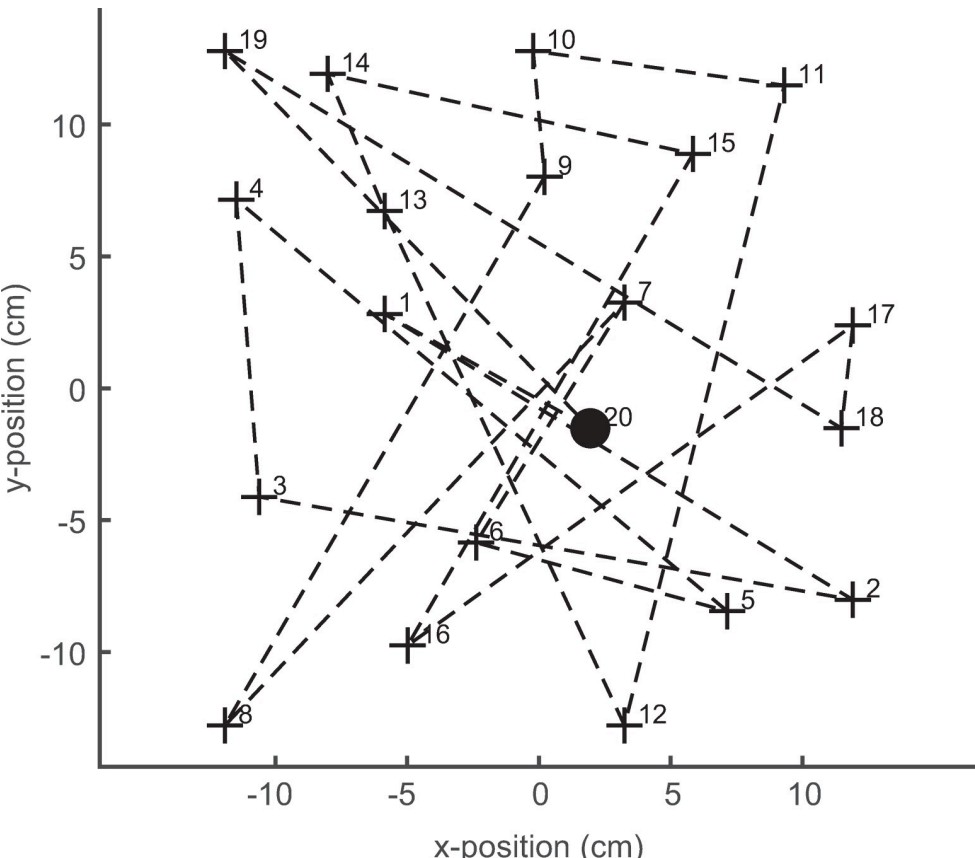

**Fig 2. Illustration of the target sequence to be learned.** The start position (solid circle) was identical with the last of the 20 target positions (crosses). In relation to the image size, the crosses are shown here twice as large as in reality. Each target was shown alone on the homogeneous background at an inter-target interval of 1.2 s.

the 20 targets of the sequence, each with a presentation time of 1.2 s. Each target appeared simultaneously with the disappearance of its predecessor. During the sequence presentation, subjects placed their hand on the table next to the graphic tablet and were only allowed to fol-low the target with their eyes.

**Reproduction phase.**   The disappearance of the last target served as the go-signal for the reproduction. For the reproduction, subjects were instructed to point on the homogenous gray screen to the target locations in the same order as they were shown during the sequence presentation. The pointing movements were triggered internally (without requiring fast move-ments). The average pointing interval of the patients (1.85±0.46 s; N = 13) was only marginally (T(25) = 2.05; p = 0.051) longer than that of the controls (1.54±0.32 s; N = 14). Subjects were not to guess and only to point to the locations they clearly remembered. Any noticed error or omission was not to be corrected. A button had to be pressed when no further target locations could be recalled. After the button press, the next training trial started with the presentation of the very same sequence.

## Study design

Three sessions with the same sequence were performed: initial training (Session 1) with 25 tri-als, a second retraining (Session 2) with only 6 trials which started 30 min after the end of the initial training, and a third one (Session 3) with 25 trials performed one week after the initial training. Three subjects (one control, two patients) were not able to attend the third session. Consequently, all analyses concerning the initial training were performed with the data of 27 participants (14 controls, 13 patients), and analyses concerning the retention after 30 min and after one week only with 24 participants (13 controls, 11 patients).

The duration of Session 1 was 22.0±3.0 min (N = 27) and did not differ between patients and controls. The patients performed the sessions with DILSS long time (10±6 yrs; see Table 1) after surgery.

An important feature of the training is that the number of targets presented before each recall clearly exceeds the capacity of the working memory. To extend the sequence stored in the long-term memory, it is therefore most efficient to keep the few targets that follow the end of the known sequence in the working memory as long as possible. This can only be achieved if the working memory is protected against overflow by the following targets. In that way, the task encourages explicit learning because it requires the subjects to actively select the targets learned in the current trial from a large amount of available information.

Both patients and controls performed the Corsi block-tapping task [30] immediately before the initial DILSS training. The Corsi task evaluates the capacity of spatial working memory. Subjects reproduced in immediate recall a finger-tapping sequence presented by the examiner at an inter-target interval of about 1 s. Pointing was performed on 9 cubes mounted on a rect-angular support plate. Each tapping sequence was presented only once. Starting with a sequence length of two, the sequence was extended by one target on the next attempt if suc-cessful. After a failure, another tapping sequence with the same length was tested. The task ter-minated after two successive failures. The task was performed twice, with forward and backward ordered recall. The *Corsi block span* was defined as the mean of the length of longest correctly reproduced sequence, averaged across the forward and the backward task.

## Clinical testing

To relate the performance of the patients in our test to their memory performance assessed clinically, the results of the CVLT and the ROCFT in the context of their neuropsychological examination are also reported here. The CVLT was evaluated in 11, and the ROCFT in 10 of

the 13 patients. In the CVLT, a sequence of 16 words is verbally presented to the subjects who recall as many of these words as possible in any order (free recall) immediately after the presentation. During the training period, verbal presentation and recall are repeated 5 times. After a period of 20 min, during which the subjects perform other neuropsychological tests, the remembered words are again tested in free recall. From this test, we analyzed the number of correctly recalled words in the last test before the 20 min retention interval (the so called *Short-delay free recall*) and in the test after the 20 min retention (*Long-delay free recall*). In the ROCFT, subjects are required to copy a complex line drawing (see e.g. [16]). Here, we report only the score (range: 0–36) patients achieved in redrawing the figure from memory, immediately after completion of the copy.

The neuropsychological examinations were performed partly before (8 patients) and partly after (3 patients) surgery. The average time between both was 54 days. Therefore, the conclusions drawn from a comparison with our test, which was performed long time (10 yrs) after surgery, are limited (see discussion).

Furthermore, all patients performed the Montreal Cognitive Assessment (MoCA) test and the Beck-Depression-Inventory test.

## Data analysis

**Segregation between reproductions and erroneous pointing movements.** Each pointing position occurring between the lift of the stylus from the starting position and the button press indicating the trial end was considered as a potential reproduction of one of the 20 target positions of the sequence. The start position (target and stylus) was excluded from the analysis. Assigning reproductions to targets under the conditions of the present study is not trivial because the random target path caused a dissociation between spatial distance and order distance. Moreover, in the absence of any visual reference except the frame of the monitor, pointing never reached high accuracy. In addition, omissions and erroneous pointing movements which do not correspond to any remembered target position may occur (*explorations*). Therefore, in this paradigm, the reproductions cannot simply be assigned to the targets with the smallest spatial distance, but were submitted to an algorithm which was developed to assign pointing positions to target locations under these specific conditions [19]. Basically, the algorithm performs a recursive ordered assignment [31, 32] searching for the longest continuous target sub-sequence that can be assigned in order and under consideration of potential intermittent, explorative reproductions which were not assigned to any target. After removing the assigned targets and reproductions from the original sequences, the algorithm restarts again until no further assignments are possible. In this way, the algorithm can account for omissions, explorations, and order errors.

**Dependent variables and statistics.** Because order errors and explorations occurred only exceptionally in the present study (order errors: 2.5%; exploration errors: 2.7%), recall performance was assessed by the number of targets assigned. The learning progress was evaluated as the time course of the recall probability across trials, computed as the fraction of the number of assigned targets with respect to the number of targets per trial (20). Since these recall probabilities are subject to noise, its time course was approximated in each individual by a cumulative normal distribution with mean and standard deviation as fitted by a generalized linear regression with a probit link-function for the dependence of the recall probability on the trial number and the assumption of a binomial distribution of each recall probability. Fig 3 shows the resulting probit-fits (red line) which were used to define the dependent variables. The *initial/final recall probability* ($p\_i$/ $p\_f$) were defined as the value of the fit at the beginning/end of the session and served as a performance measure. To evaluate how fast subjects extended the

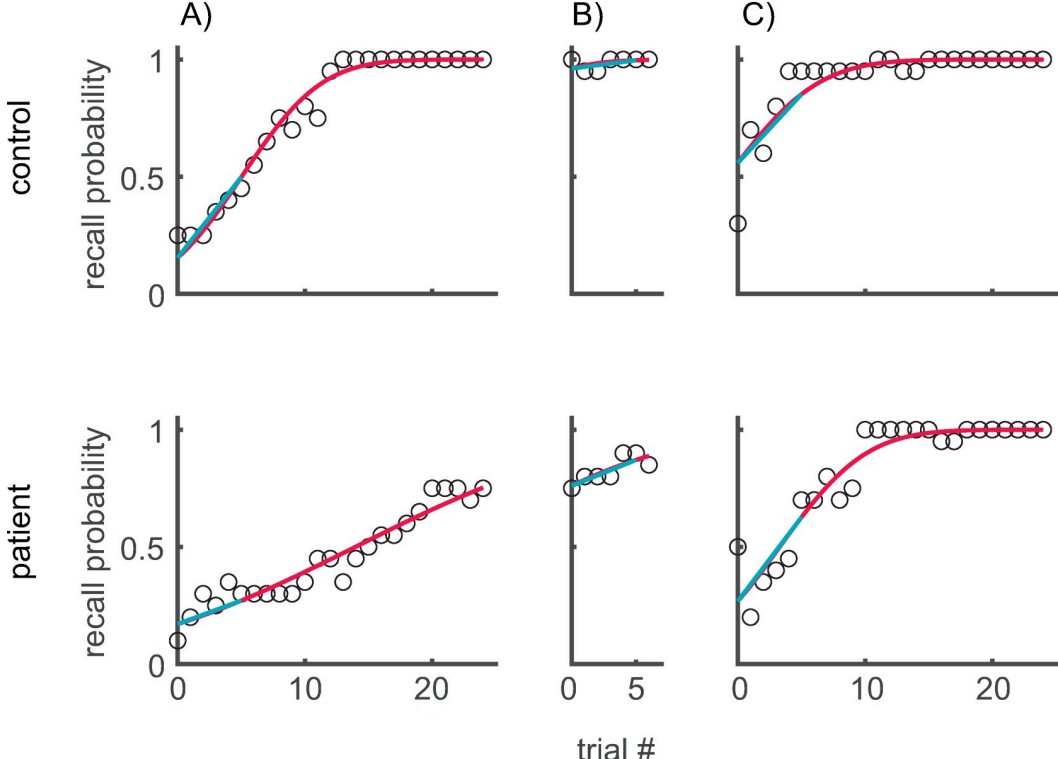

**Fig 3. Learning progress for two individuals (Top: control; Bottom: patient) during the three learning blocks.** Circles show the recall probability, defined as the fraction of the number of recalled targets with respect to the number of presented targets (i.e. the sequence length N = 20) for each reproduction (trial #). Red line: The approximation of the recall probability by a generalized regression (probit-fit). Cyan: The line starting at the initial recall probability and the slope equal to the mean learning progress (increment of the recall probability) averaged across the first 5 trials. The slope of this line is used to define the initial learning speed. A) initial training, B) retention test starting 30 min after the end of the initial training, C) training session one week after the initial training.

length of the memorized sequence, the *initial learning speed* was defined as the difference in the probit-fit between the sixth and the first trial of the session, divided by 5 and multiplied by the sequence length (20) to express the learning speed in units of targets per trial.

For each subject, *retention* ($\varrho$) after 30 min and after one week was defined as the initial recall probability of Session 2 and Session 3, expressed as a fraction of the final recall probability at the end of Session 1. In the CVLT, the retention ($\varrho_{CVLT}$) was defined as the *Long-delay free recall*, expressed as a fraction of the *Short-delay free recall*.

Across subjects, the recall probabilities in DILSS were not distributed normally, especially the final recall probabilities which were close to one. Therefore, the analysis of the group effect (controls/patient right/patients left) was performed by applying parametric statistics (ANOVA, t-tests) to the logit-transformed recall probabilities. The normality of the logit-transformed probabilities was confirmed by the Lilliefors test. The parametric statistics (mean, quartiles, 95% confidence interval of the mean) of the logit-transformed recall probabilities were submitted to the inverse logit-transformation to compute the corresponding descriptive statistics of the recall probabilities (median, quartiles, 95% confidence interval of the median). These descriptive statistics are referred to by the bars and whiskers of Fig 4B, and by the group median [interquartile range] recall probabilities reported in the text. To distinguish the standard t-test and the standard ANOVA from their application on the logit-transformed dependent variables, the latter are referred to here as the 'logit t-test' and as the 'logit ANOVA'.

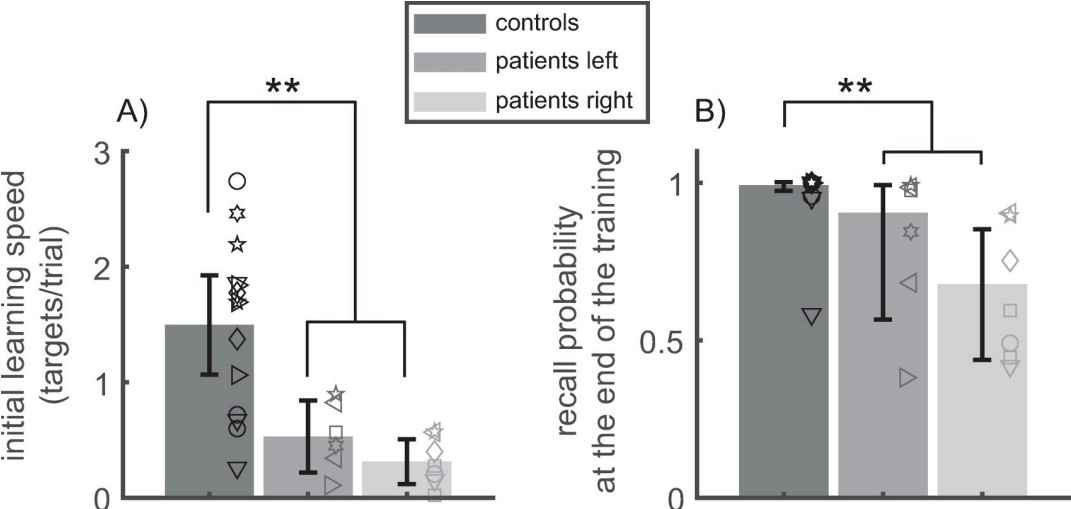

**Fig 4. Learning progress during the initial training (Session 1) with DILSS.** A) *initial learning speed* (targets/trial). Each symbol indicates the slope of the cyan line in Fig 3A, multiplied by the total sequence length (20) for one subject. B) *final recall probability* at the end of Session 1. Symbols show individual values. Bars: median; Whiskers: 95% confidence interval of the median. Both learning speed and final recall probability were clearly impaired (solid double asterisk: p<0.001) in patients.

The distribution of the DILSS-retention ($\varrho$), the *Short-delay free recall* in the CVLT, the CVLT-retention ($\varrho_{CVLT}$), as well as the distribution of the Corsi block span did not significantly differ from a normal distribution. This was confirmed with the Lilliefors test. Therefore, group effects on these dependent variables were tested using ANOVA and standard (paired or unpaired) t-tests. In the text, their descriptive statistics are reported by mean±standard deviation.

Pearson's correlation coefficient ($\rho$) was calculated after eliminating outliers that fell outside the 2D 95% confidence ellipse (see Fig 7). The false-positive probability of the empirical $\rho$ was calculated by the t-test applied to the Student distributed random variable $t = \rho \cdot \sqrt{\frac{N-2}{1-\rho^2}}$.

Effects with false positive probabilities α<0.01 are considered highly significant (**), α<0.05 as significant (*), and α<0.1 as a trend. All t-tests were two-tailed.

Whenever a dependent variable differed significantly (p<0.05) between controls and patients, or between patients with right-sided and left-sided lesions, we evaluated the performance of a binary classifier in correctly assigning subjects to one or the other group, based on that dependent variable. This was done by means of an ROC-analysis [33] by computing the false-positive rate (*FPR*) and the true positive rate (*TPR*) as a function of the decision threshold (*t*) in the dependent variable. The optimal threshold ($t_{opt}$) was defined as the one maximizing the correct-decision rate (CDR) for equal probability of both classes:

$$CDR(t) = \frac{1 - FPR(t) + TPR(t)}{2}$$

$$t_{opt} = \arg\max_t CDR(t); \quad CDR_{opt} = CDR(t_{opt})$$

## Results

All individual test results are summarized in S1 Table. The MoCA-test showed normal values (>26) for all participants. The Beck-depression Inventory showed in general low scores below

8. In only four subjects (two patients as well as two controls) was the score 10 to 12, still well below the cutoff of 14 for clinically symptomatic values [34]. None needed specific therapy.

## Analysis of the initial training session with DILSS

**Initial recall probability.** The initial recall probability immediately after the very first presentation of the sequence did not differ (ANOVA: $F(2,24) = 0.14$; $p = 0.87$) between controls, patients with left TLE, and patients with right TLE. The global mean was $p\_i = 0.21\pm0.14$ ($N = 27$). By multiplication with the number of targets (20) one obtains the corresponding number of reproductions ($4.24\pm2.87$) achieved in immediate recall.

**Initial learning speed.** Fig 4A shows the initial learning speed in the first training with DILSS (Session 1). On average, the *initial learning speed* of patients ($0.41\pm0.27$ targets/trial) was significantly smaller ($T(25) = 4.95$; $p<0.0001$) and less than one third of that of the controls ($1.50\pm0.74$ targets/trial). Using an optimal threshold of $t_{opt} = 0.58$ targets/trial subjects could be classified as patients and controls with an optimal correct-decision rate of $CDR_{opt} = 0.89$.

The *initial learning speed* of patients with right-sided lesions ($0.31\pm0.21$ targets/trial) tended only weakly ($T(11) = 1.55$; $p = 0.15$) to be slower than that of patients with left-sided lesions ($0.53\pm0.30$ targets/trial).

**Final recall probability.** The impairment of the patients in spatial sequence learning was also reflected by the *final recall probability* at the end of Session 1 (Fig 4B). The *final recall probability* was smaller (logit t-test: $T(25) = 4.89$; $p<0.0001$) in patients (median [interquartile range] = 0.81 [0.35]) than in controls (0.99 [0.03]). This reflects the fact that, even though patients showed considerable learning progress, none of them could reproduce the entire sequence at the end of the initial training session as most of the controls could (see column $p\_f$ in S1 Table). The Spearman's rank correlation coefficient between the *final recall probability* and the initial learning speed was close to one (0.98). Consequently, $CDR_{opt} = 0.893$ of the *final recall probability* was similar to that of the *initial learning speed*. The optimal decision threshold was $t_{opt} = 0.993$.

The *final recall probability* tended ($T(11) = 1.78$; $p = 0.10$) to be more strongly impaired in patients with right-sided lesions (0.68 [0.31]) than in those with left-sided lesions (0.90 [0.25]).

In summary, both *initial learning speed* and *final recall probability* quantify the ability to acquire sequential visuospatial memory in DILSS. This ability was impaired in TLE-patients with both left-sided and right sided lesions and the impairment tended to be stronger for right-sided lesions.

## Analysis of long-term retention in DILSS

Fig 5 shows the *retention* after 30 min and after one week, defined as the initial recall probability in Sessions 2 and 3, expressed as a fraction of the final recall probability in Session 1. After 30min, retention was almost perfect in all three groups (controls: $0.97\pm0.06$; patients left: $\varrho = 0.95\pm0.09$; patients right: $\varrho = 0.87\pm0.14$). The retentions after 30 min did not significantly differ from one (t-tests: $p>0.08$) and also not from each other (ANOVA: $F(2,21) = 2.61$; $p = 0.097$). In contrast, the long-term retention after one week differed significantly between groups (ANOVA: $F(2,21) = 4.08$; $p = 0.031$). It was significantly smaller than one for controls ($\varrho = 0.88\pm0.14$, $p<0.02$) and for patients with left-sided lesions ($\varrho = 0.55\pm0.28$, $p<0.02$) but not for patients with right-sided lesions ($0.84\pm0.38$), due to large inter-subject differences. These results indicate that systematic impairments of long-term retention of the sequential visuospatial memory were confined to the TLE-patients with left-sided lesions. This side asymmetry was contrary to that of the *final recall probability* at the end of Session 1, which was more strongly impaired in right-sided lesions. This

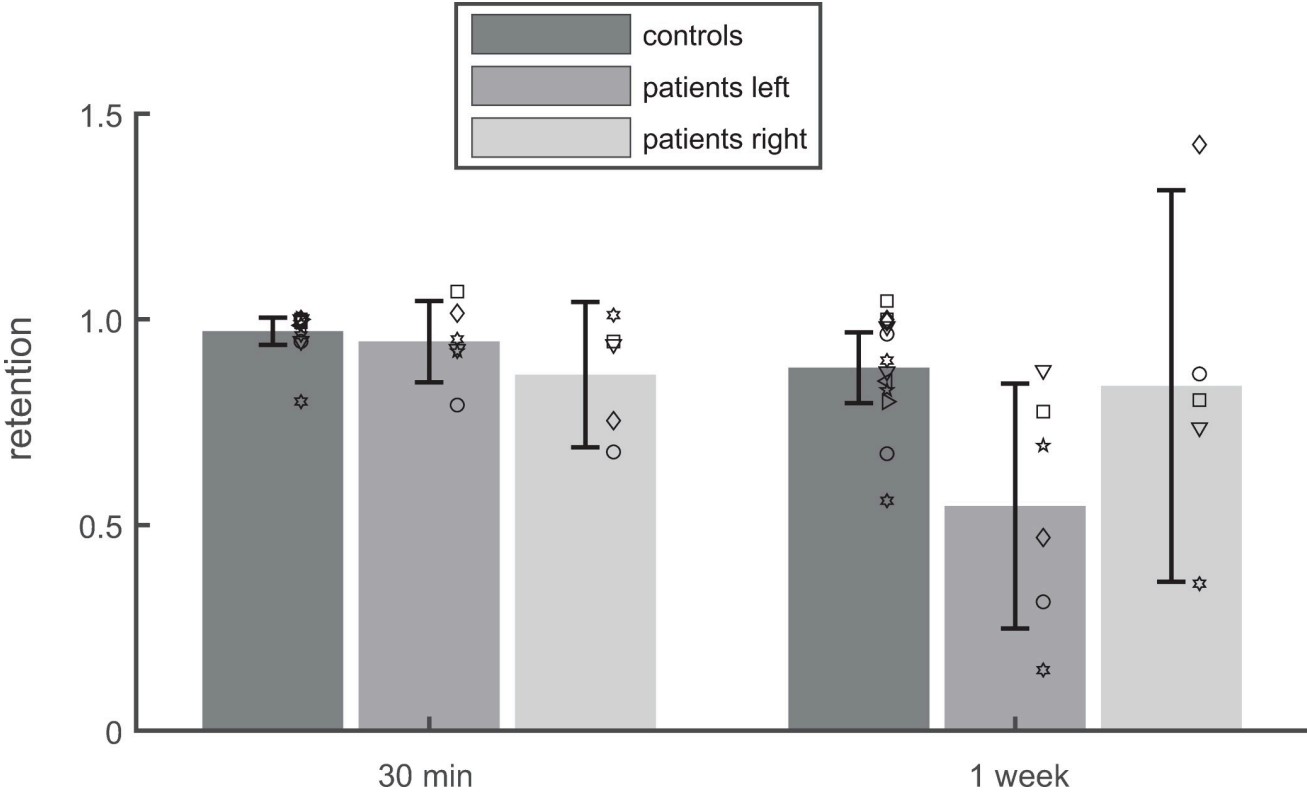

**Fig 5. Retention in DILSS.** Bars and whiskers show the mean and the 95% confidence interval of the mean of the *retention* ($\varrho$) after 30 min and after one week; Symbols: retention of individuals. TLE-patients with lesions on the left side show smaller retention than the controls after one week but not after 30 min.

is summarized in Table 2: Left-sided patients learned almost as well as the controls but forgot more than the controls after 1 week, while right-sided patients show impaired initial learning combined with good retention after one week.

## Spatial memory span

Fig 6 shows that the Corsi block span was larger in controls (6.54±0.80) than in patients (5.77 ±0.75; t-test: T(25) = 2.57; p = 0.017) and did not differ between patients with lesions on the right and on the left side. However, all patients preserved considerable spatial working

**Table 2. Performance summary of memory acquisition and retention in DILSS.**

| Group\Condition | Immediate recall probability [a] | Trained recall probability [b] | Retention $\varrho$ [c] (30 min) | Retention $\varrho$ [c] (1 week) | Retention loss $\Delta\varrho$ [d] |
|---|---|---|---|---|---|
| Controls | 0.22±0.18 | 0.99 [0.03] | 0.97±0.06 | 0.88±0.14 | 0.09±0.14 |
| Patients left | 0.18±0.10 | 0.90 [0.25] | 0.95±0.09 | **0.55±0.28** | **0.40±0.27** |
| Patients right | 0.21±0.10 | **0.68 [0.31]** | 0.87±0.14 | 0.84±0.38 | 0.03±0.49 |
| Patients all | 0.20±0.10 | 0.81 [0.35] | 0.91±0.12 | 0.68±0.35 | 0.23±0.41 |

The numbers printed in bold indicate the patient group with the strongest impairment of the trained recall and of the long-term retention.

[a] Mean ± standard deviation of the initial recall probability ($p\_i$) after the first sequence presentation in Session 1.

[b] Median [interquartile range] of the final recall probability ($p\_f$) at the end of Session 1.

[c] Mean ± standard deviation of the retention, defined as the initial recall probability ($p\_i$) at the beginning of Session 2 (30 min) and Session 3 (1 week), expressed as a fraction of $p\_f$.

[d] Mean ± standard deviation of the paired difference of the retention between 30 min and one week.

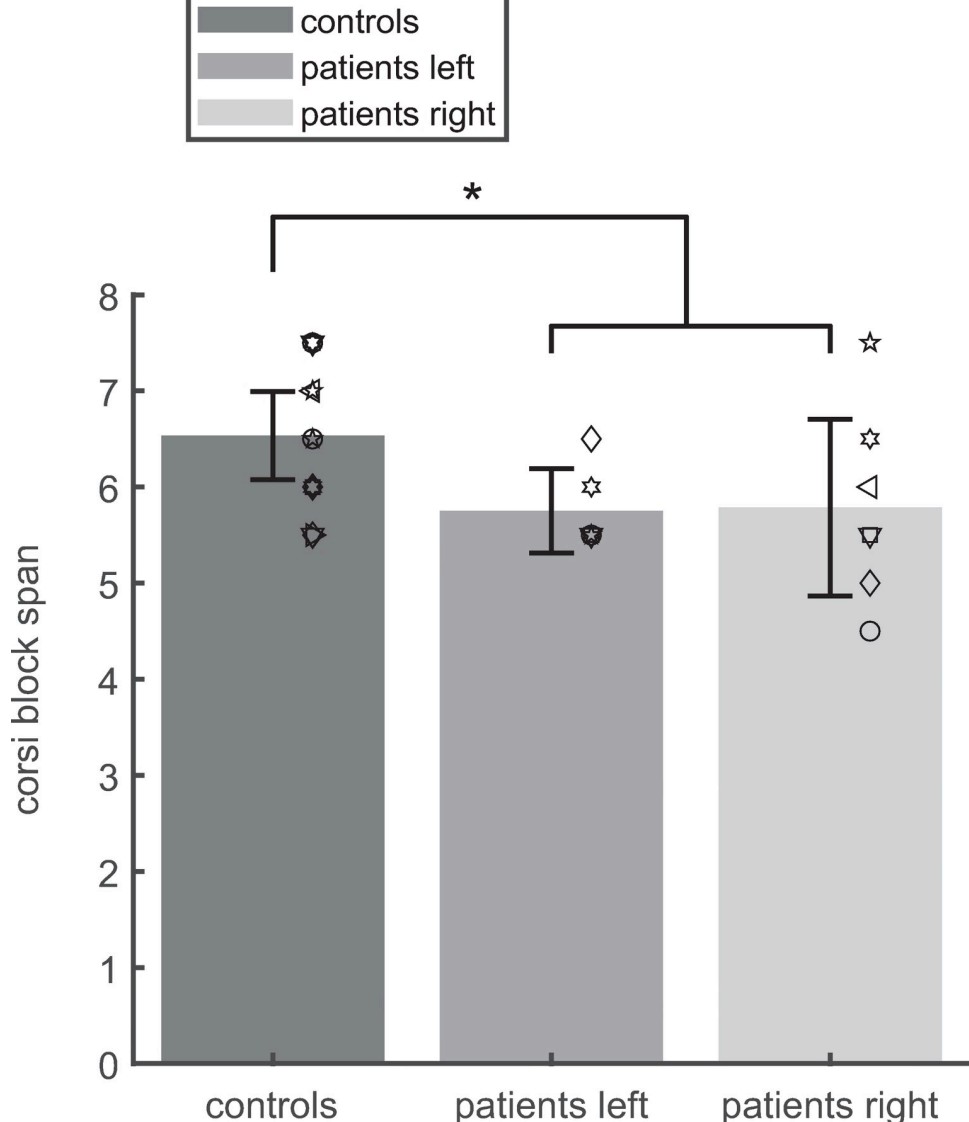

**Fig 6. Mean (bars) and the 95%-confidence interval of the mean (whiskers) of the working-memory span as evaluated in the Corsi block-task.** Symbols: individual subjects. The patients' memory span was impaired (solid asterisk: p = 0.012) with respect to controls.

memory performance. The lowest Corsi block span was 4.5 (see S1 Table). For the controls, the spatial memory span (min: 5.5; max: 7.5) was not critical for the memory acquisition in DILSS as the Corsi block span did not correlate (Pearson's $\rho$ = -0.0436; T(12) = -0.1511, p = 0.89) with the *initial learning speed* in Session 1. In the patients, these two variables showed a moderate positive correlation Pearson's rho = 0.57; T (11) = 2.27, p = 0.04). Classification between TLE-patients and controls based on a lower limit of the Corsi span for the controls ($t_{opt}$ = 5.5) was relatively poor ($CDR_{opt}$ = 0.70).

## Word list learning

The patients' performance in the CVLT (*Short-delay free recall*: 12.55±1.92, N = 11) was in the range of or even better than the standard values reported in the literature [9] showing that our

patients were not impaired in verbal learning. However, the *Short-delay free recall* of patients with left-sided lesions (11.20±1.92, N = 5) was significantly (T(9) = -2.72; p = 0.023) smaller than that of patients with right-sided lesions (13.67±1.03, N = 6). The *Short-delay free recall* did not differ (T(9) = 0.92; p = 0.38) between patients tested before or after surgery. The ROC-analysis showed that the *Short-delay free recall* could efficiently predict the lesion side. Classifying all patients with a *Short-delay free recall* above $t_{opt}$ = 13 as right-sided lesions and all those below that threshold as left-sided lesions, resulted in a correct-decision rate of $CDR_{opt}$ = 0.83.

The CVLT-retention twenty minutes after training ($\varrho_{CVLT}$ = 1.03±0.13, N = 11) did not differ from one (T(10) = 0.76; p = 0.47) and did also not differ between right-sided and left-sided lesions (T(9) = -0.98; p = 0.35). This shows that the verbal memory of the patients did not show any retention loss across the 20 min delay period or a dependence of this retention loss on the lesion side.

Comparison of verbal and visuospatial memory

The above analysis of right-left asymmetries of the recall in CVLT and DILSS revealed that patients exhibited opposite trends. In the CVLT, patients with lesions on the left side performed worse than those with lesions on the right, whereas the opposite happened in DILSS. To investigate the relation between the performances in CVLT and DILSS in the TLE-patients in more detail, we correlated the trained recall in both tasks with each other. To express recall performance in both tasks as a relative measure with respect to optimal performance, we divided the number of correctly recalled words in the CVLT by the total number of presented words (16), thus obtaining an estimate of verbal recall probability. This normalization was performed for both Short-delay free recall and Long-delay free recall. The results are shown in Fig 7.

Immediately after the training phase (Fig 7A), the *final recall probability* in DILSS correlated negatively with the normalized *Short-delay free recall* in CVLT (ρ = -0.71; T(8) = -2.83; p = 0.02) indicating that patients with better performance in CVLT performed worse in DILSS and vice versa. The retention ($\varrho$, $\varrho_{CVLT}$) in the delayed recall (30 min after the end of the training in DILSS, and 20 min after training in the CVLT) did not correlate between the tasks (ρ = -0.07; T(9) = -0.21; p = 0.84).

### Immediate recall in the ROCFT

In the immediate recall of the ROCFT, patients obtained on average a score of 19.55±4.38 (N = 10) which did not differ significantly (T(125) = 0.26; p = 0.80) from that of a healthy population (20.08±6.35; N = 117 [35]). Discrimination between patients and controls based on ROCFT was almost random, as the optimal rate of correct decisions for these distributions was only $CDR_{opt}$ = 0.56.

### Discussion

The results of the current study reveal two main aspects. First, temporal lobe lesions in TLE patients caused impaired learning speed and trained recall in the DILSS task. Second, these impairments tended to be more pronounced in right-sided lesions than in left-sided lesions. Both initial learning speed and the final recall probability after training with DILSS were informative parameters to classify subjects as patients or controls with an optimal correct-decision rate of $CDR_{opt}$ = 0.89. Thus, the DILSS-tasks shows high sensitivity and specificity in discriminating between controls and resected TLE-patients. This supports the motivating hypothesis of the current study that DILSS provides an efficient and specific test procedure for the acquisition of episodic, sequential visuospatial memory. The second aspect of the results, i.e., the observed hemispheric asymmetry, had the same direction as observed by Tucker, Novelly [18]

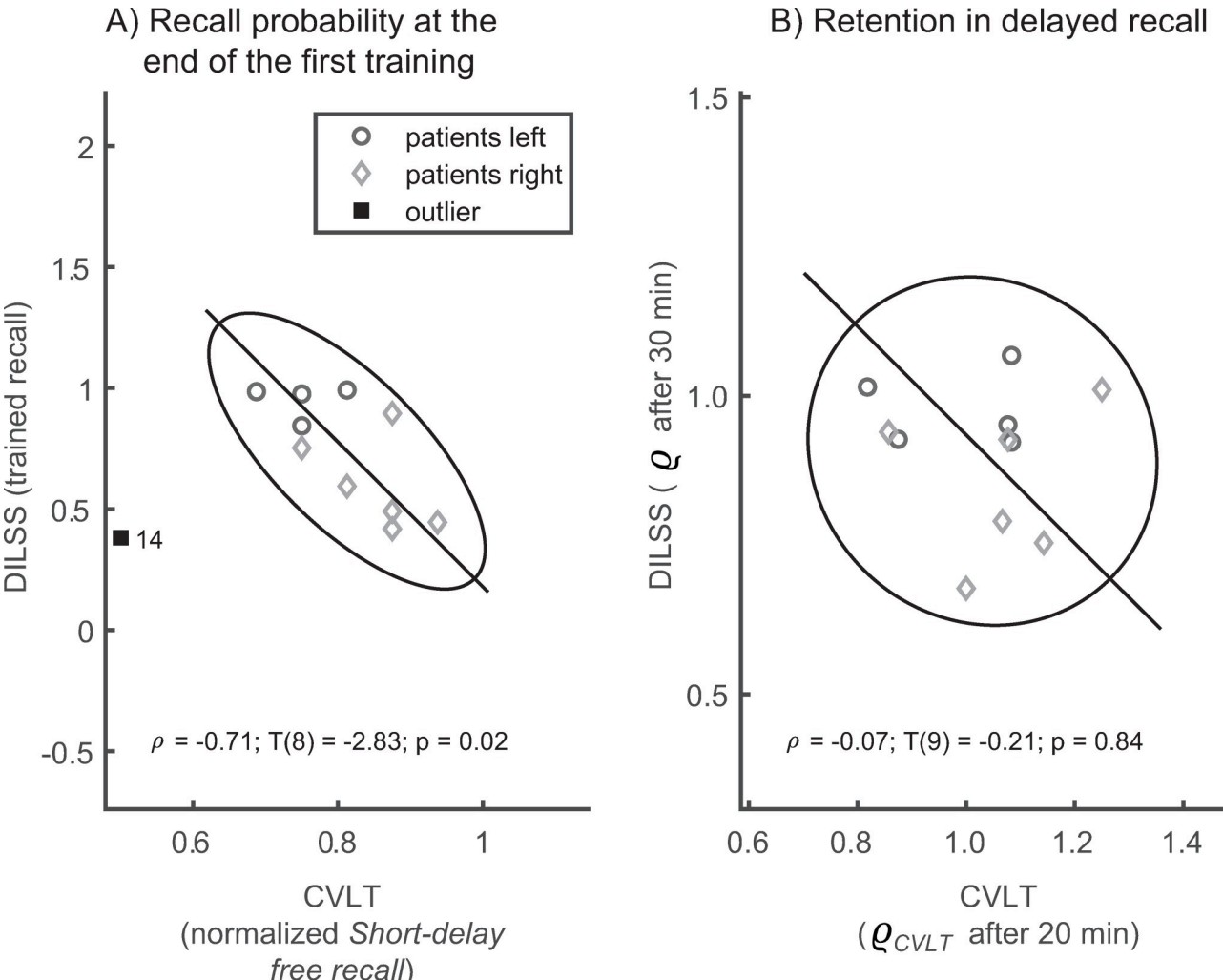

**Fig 7.** A) In TLE-patients, the initial recall probabilities of verbal memory in CVLT were negatively correlated with the recall probabilities of spatial sequence learning in DILSS immediately after training. B) The retention ($\varrho$, $\varrho_{CVLT}$), defined as the delayed recall expressed as a fraction of trained recall, did not correlate between CVLT and DILSS. To visualize the correlation, all axes are scaled to the standard deviations of the respective variable. The ellipse shows the 95% confidence range and the solid line its major axis. The solid symbols show the outlier and the number printed nearby indicates the corresponding patient ID (see Table 1 and S1 Table).

after training with the sequential presentation mode. Both experiments are also consistent in that no such asymmetry was observed in immediate recall ($p\_i$ in Session 1). Thus, in line with the study of Tucker, Novelly [18], our study provides further evidence that the right temporal lobe (and the right hippocampus) plays a specific role in the buildup of long visuospatial sequences.

Neither in the verbal (CVLT), nor in the visual (ROCFT) memory test were our TLE-patients generally impaired compared to healthy controls. The good performance of the patients in these tests reflects a positive bias, since the patients recruited from the outpatient seizure clinic were all highly motivated and showed a personal interest in the study. The overall good performance of our patients is also reflected in their good post-operative outcome (Engel Epilepsy Surgery Outcomes), their normal psychiatric status, and their fulltime employment. In contrast, these patients showed a marked impairment in our episodic visuospatial memory task. Even though the comparability between the tasks is limited because the DILSS was

performed long after and the CVLT and ROCFT mainly before surgery, this difference does not seem to explain the different results of the test procedures for two reasons. First, the long recovery period of the patients before the DILSS test would explain a lower sensitivity and not the higher one actually observed. Second, previous studies showed that the performance of patients with epilepsy in verbal memory tests is not significantly affected by insulectomy [36] or by temporal lobe resection [37]. The study of Lendt, Helmstaedter [37] showed even a small trend of improvement after surgery. Similarly, the deficits of TLE-patients in a visual memory test (DCS) were not affected by amgydalohippocampectomy [38]. This is compatible with the hypothesis that the removed tissue is dysfunctional due its pathology. The impairment of the DILSS in the patients of the current study therefore suggests that this test reveals permanent deficits in episodic visuospatial memory to which the CVLT or the ROCFT were less sensitive.

The observation that the side asymmetries in DILSS were opposite to those of the CVLT confirms the hypothesis that DILSS challenges primarily visuospatial memory differently to the CVLT. However, initial learning speed was significantly impaired in both right and left lesions (Fig 4A), pointing to an important contribution of both sides to this task. Further studies are necessary to investigate potential differences between DILSS and other non-sequential visuospatial memory tests such as the ROCFT or the DCS. An obvious hypothesis is that the inferotemporal cortex, which plays an essential role in object recognition [39], also plays a greater role in visual gestalt-based memory, and thus in performance in the ROCFT, than in sequential visual memory or in DILSS.

The interpretation of the retention observed in the current study is complicated by the fact that our training design did not attempt to match the initial learning performance between patients and controls [27, 40]. Under these conditions, recall performance after a given retention interval (30 min or one week) would not only reflect the stability of the acquired memory but also the performance at the end of the training. Therefore, we defined our retention measure $\varrho$ as the recall performance after a retention interval, expressed as a fraction of the recall performance at the end of the initial training ($p\_f$ in Session 1). Pearson's coefficient of correlation between the initial learning performance ($p\_f$ in Session 1) and the retention measure $\varrho$ after one week was only r = 0.09 and did not significantly (T(22) = -0.43; p = 0.67) differ from zero. This suggests that our normalization procedure was successful in that $\varrho$, as intended, did not show a linear dependence on the initial learning. However, the normalization cannot exclude confounding of retention and learning as efficiently as matching can. With these caveats, the results concerning the retention $\varrho$ suggest that long-term retention was impaired in patients with left-sided lesions (Fig 5). Such an impairment of long-term retention resembles the findings of Visser, Forn [27] who reported the same effect in long-term retention in the Rey Auditory Verbal Learning Test [41]. On the group level in patients with right-sided lesions the long-term retention ($\varrho$ = 0.84) in DILSS did not differ from one (Fig 4A). However, because of the large inter-subject variance, this does not necessarily indicate that their retention was unimpaired. In the verbal memory task, Visser, Forn [27] did not observe a side asymmetry in long-term retention of verbal information.

Obviously, performance in DILSS depends on working memory, since subjects must concentrate on the subsequence to be learned in the current trial. Nevertheless, the speed of the extension of the memorized sequence did not correlate with the spatial working memory span in healthy subjects. Thus, the specific ability to extend the memorized sequence in DILSS does not just reflect working memory performance but can be used to measure the transfer of spatial information to the long-term memory. The correlation of working memory span with the DILSS speed in the patients may indicate that the lesioned temporal areas play a role in both visuospatial working memory and visuospatial long-term memory.

The clinical relevance of the presented test was shown by the high correct decision rate of about 0.9 when classifying between controls and patients based on the DILSS speed. This is a major advantage in comparison to the CVLT or the ROCFT, in which our patients were not impaired compared to healthy controls. The data presented here suggest that the DILSS could complement the CVLT.

## Limitations

The results of the current study are not sufficient to validate the DILSS test presented here for routine clinical use. The number of participants was too small for such a claim, and the matching between controls and patients for other factors such as age was not well enough controlled. Memory performance generally deteriorates with age as has been reported by many studies for verbal memory [42, 43] and also for the Rey-Osterieth Complex Figure Test (ROCFT) [35]. These studies generally show a significant decline in performance above the age of 50 but relatively stable performance between ages 20 and 40. Therefore, the age difference between our patients (age 40.2±11.1) and controls (29.5±9.7 yrs) does not explain the highly significant (p<0.0001) difference in learning speed we observed in DILSS. The current study, despite its limitations in terms of group matching and sample size, suggests that DILSS is a promising test to detect deficits in episodic visuospatial memory. To validate the test for clinical use, more extensive studies are needed in which the new test is applied in a larger population and under more stringent selection criteria.

## Conclusion

In summary, DILSS offers a possibility to test the acquisition of episodic, sequential visuospatial memory. In TLE-patients, the test proved to be more sensitive than the CVLT or the ROCFT in distinguishing patients and controls. In DILSS, patients with right-sided lesions were more strongly impaired than those with left-sided lesions, in agreement with the general hypothesis of right-hemispheric dominance in visuospatial processing.

## Supporting information

**S1 Table. Individual performances in DILSS, Corsi block-task, and neuropsychological examination.**
(PDF)

## Acknowledgments

We thank J. Remi, and F. Filippopulos for technical assistance with patients and K. Göttlinger for copyediting the manuscript.

## Author Contributions

**Conceptualization:** Thomas Eggert, Andreas Straube.

**Data curation:** Phuong Van Nguyen.

**Formal analysis:** Thomas Eggert.

**Investigation:** Phuong Van Nguyen.

**Methodology:** Thomas Eggert.

**Project administration:** Andreas Straube.

**Resources:** Katharina Ernst, Sandra V. Loosli.

**Software:** Thomas Eggert.

**Writing – original draft:** Thomas Eggert.

**Writing – review & editing:** Thomas Eggert, Katharina Ernst, Sandra V. Loosli, Andreas Straube.

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
