## [Decision Letter · Decision Letter 0]

8 Jun 2022

PONE-D-22-14606A new test to detect impairments of sequential visuospatial memory due to lesions of the temporal lobePLOS ONE

Dear Dr. Eggert,

Thank you for submitting your manuscript to PLOS ONE. After careful consideration, we feel that it has merit but does not fully meet PLOS ONE’s publication criteria as it currently stands. Therefore, we invite you to submit a revised version of the manuscript that addresses the points raised during the review process.

We look forward to receiving your revised manuscript.

Kind regards,

Daichi Sone

Academic Editor

PLOS ONE

Journal Requirements:

3. We note that you have referenced (ie. Bewick et al. [5]) which has currently not yet been accepted for publication. Please remove this from your References and amend this to state in the body of your manuscript: (ie “Bewick et al. [Unpublished]”) as detailed online in our guide for authors

Additional Editor Comments:

I echo the reviewers' comments. Particularly, the distinct group-difference in age and possible cognitive decline at the presurgical stage should be appropriately addressed, although the topic of visual episodic memory in TLE is interesting.

Reviewers' comments:

Reviewer's Responses to Questions

**Comments to the Author**

1. Is the manuscript technically sound, and do the data support the conclusions?

Reviewer #1: Yes

Reviewer #2: Partly

2. Has the statistical analysis been performed appropriately and rigorously? 

Reviewer #1: Yes

Reviewer #2: N/A

3. Have the authors made all data underlying the findings in their manuscript fully available?

Reviewer #1: Yes

Reviewer #2: Yes

4. Is the manuscript presented in an intelligible fashion and written in standard English?

Reviewer #1: Yes

Reviewer #2: Yes

5. Review Comments to the Author

Reviewer #1: The authors evaluated visuospatial episodic memory of patients with epilepsy after temporal lobo resection and healthy controls using sequential visuospatial memory test. This new neuropsychological test showed impaired visuospatial memory in patients after right side resection in the first evaluation, but retention rate at one week later was worse in patients after left side resection. The standard neuropsychological tests failed to show the difference. The authors focused on visuospatial memory function which is sometimes undervalued in the perioperative workup and successfully demonstrated the left-right difference of the hippocampal function using a novel sequential test developed to specifically extract episodic memory. This study is valuable and is worth publishing in PLO SONE, if the following concerns are appropriately addressed.

Methods

L130.　Please clarify the inclusion criteria of the patients after epilepsy surgery. Did you include all patients who underwent epilepsy surgery, became seizure free, and agreed to join the study?

L133.　Typo? The information on seizure onset is provided in S1, not S2. Anyway, the demographic data shown in S1 is important to interpret the results. S1 should be in the main text, not in the supplementary.

L135.　Was the language dominance determined solely by handedness? If so, uncertainty in the language dominance should be mentioned as an unignorable limitation of this study.

L201.　The phrase "training sessions" implies that there will be a test session. This is just a comment.

Discussion

L497.　The authors wrote, "due to TLE resection". But, it would be inappropriate to attribute the memory impairment solely to resection, as it would had originally been impaired due to prolonged epilepsy.

S1.　According to the table, patient 6 and 9 underwent tumor resection. Were their hippocampi resected? Please clarify whether the hippocampus of each patient was removed or not.

Fig1.　It is unclear whether the hippocampus of each patient was removed from the slices in this figure. For example, the hippocampus of patient 6 appears to remain. Mostly, the slices seem too close to the middle base. The petrosal bone should not be in the slice in order to show the hippocampus correctly.

Reviewer #2: Thomas Eggert al. applied the DILSS-task, a new method to detect impairments of sequential visuospatial memory in 13 patients with temporal lobe epilepsy and 14 healthy controls. The result indicated that learning speed was reduced in patients, also right-sided lesions induced worse impairments than left-sided lesions, which show the right hemispheric dominance in visuospatial processing.

Limitations:

1) Treatments, age would influence the results and hence should be controlled. In line 130, the average age of the control group was 29.5 ± 9.7, and that of the patient group was 40.2 ± 11.1. The age factor was not controlled; In addition, the treatment plan and effect before and after right temporal lobectomy and left temporal lobectomy were not specified in detail. This factor may have an impact on the evaluation of patients' memory function.

2）There is no indication whether the test was conducted by the same observer, and there may be observer bias.

3）There were only 13 patients and 14 control in the groups, the sample size was too small to support the result.

Other comments:

"CVL" should be changed to "CVLT" in line 513.

6. PLOS authors have the option to publish the peer review history of their article (what does this mean?). If published, this will include your full peer review and any attached files.

Reviewer #1: No

Reviewer #2: No

---

## [Author Response · Author response to Decision Letter 0]

24 Jun 2022

The responsees to the reviewers are included in a separate uploaded file ('ResponseLetter.docx')

---

## [Decision Letter · Decision Letter 1]

19 Jul 2022

A new test to detect impairments of sequential visuospatial memory due to lesions of the temporal lobe

PONE-D-22-14606R1

Dear Dr. Eggert,

We’re pleased to inform you that your manuscript has been judged scientifically suitable for publication and will be formally accepted for publication once it meets all outstanding technical requirements.

Kind regards,

Daichi Sone

Academic Editor

PLOS ONE

Additional Editor Comments (optional):

Reviewers' comments:

Reviewer's Responses to Questions

**Comments to the Author**

1. If the authors have adequately addressed your comments raised in a previous round of review and you feel that this manuscript is now acceptable for publication, you may indicate that here to bypass the “Comments to the Author” section, enter your conflict of interest statement in the “Confidential to Editor” section, and submit your "Accept" recommendation.

Reviewer #1: All comments have been addressed

Reviewer #2: All comments have been addressed

2. Is the manuscript technically sound, and do the data support the conclusions?

Reviewer #1: Yes

Reviewer #2: Yes

3. Has the statistical analysis been performed appropriately and rigorously? 

Reviewer #1: Yes

Reviewer #2: Yes

4. Have the authors made all data underlying the findings in their manuscript fully available?

Reviewer #1: Yes

Reviewer #2: Yes

5. Is the manuscript presented in an intelligible fashion and written in standard English?

Reviewer #1: Yes

Reviewer #2: Yes

6. Review Comments to the Author

Reviewer #1: (No Response)

Reviewer #2: I have read your revision with great interest. You have certainly taken all comments of the reviewers.

7. PLOS authors have the option to publish the peer review history of their article (what does this mean?). If published, this will include your full peer review and any attached files.

Reviewer #1: No

Reviewer #2: No

---

## [Editor Report · Acceptance letter]

21 Jul 2022

PONE-D-22-14606R1 

A new test to detect impairments of sequential visuospatial memory due to lesions of the temporal lobe 

Dear Dr. Eggert:

I'm pleased to inform you that your manuscript has been deemed suitable for publication in PLOS ONE. Congratulations! Your manuscript is now with our production department. 

Kind regards, 

on behalf of

Dr. Daichi Sone 

Academic Editor

PLOS ONE